# Effect of Static and Dynamic Stretching on Corneal Fibroblast Cell

**Zhi-Xuan Dai** [1], **Po-Jen Shih** [2,*], **Jia-Yush Yen** [1,3] **and I-Jong Wang** [4]

1 Department of Mechanical Engineering, National Taiwan University, Taipei 10617, Taiwan; d07522013@ntu.edu.tw (Z.-X.D.); jyen@mail.ntust.edu.tw (J.-Y.Y.)
2 Department of Biomedical Engineering, National Taiwan University, Taipei 10617, Taiwan
3 Department of Mechanical Engineering, National Taiwan University of Science and Technology, Taipei 10607, Taiwan
4 Department of Ophthalmology, National Taiwan University Hospital, Taipei 10002, Taiwan; ijong@ms8.hinet.net
* Correspondence: pjshih@ntu.edu.tw; Tel.: +886-2-2732-1034

**Abstract:** A strain gradient was created by punching a hole in the center of a stretched elastic polydimethylsiloxane membrane to determine the effect of different strains on cultured human keratocytes (HK). In this study, two stretching methods were used: continuous stretching and cyclic stretching. Continuous stretching is relatively static, while acyclic stretching is relatively dynamic. These methods, respectively, represented the effects of high intraocular pressure and rubbing of the eyes on corneal cells. Image processing codes were developed to observe the effects of stress concentration, shear stress, continuous stretching, and cyclic stretching on HKs. The results demonstrate that stretching and shear stress are not conducive to the proliferation of corneal cells and instead cause cell death. A 10% strain had greater inhibitory effects than a 3% strain on cell proliferation. Cell survival rates for continuous stretching (static) were higher than those for cyclic stretching (dynamic). The stretching experiment revealed that cyclic stretching has a greater inhibitory effect on the growth and proliferation of corneal cells than continuous stretching. Accordingly, it shows that cyclic loading is more harmful than high intraocular pressure (static loading) to corneal cells.

**Keywords:** human keratocytes; stress distribution; cyclic stretching; continuous stretching; finite element analysis (FEA)

## 1. Introduction

Cytomechanics, an important field in biomechanics, is the study of the mechanical properties (deformation, elastic constant, viscoelasticity, and adhesion, among others) of the cell membrane and cytoskeleton under the action of load. These mechanical factors influence cell growth, development, maturity, proliferation, aging, and death [1,2].

In many cytomechanics studies, experimenters have applied mechanical stresses to cells, such as periodic stretching, pressure, shearing, or an electric field. Consequently, the strength, duration, and frequency of these mechanical forces affect cell responses [3,4]. Mechanical stretch is detected in the membrane before mechanical signals are relayed to the cortical cytoskeleton [5–7]. Then, different force systems are applied. Neidlinger-Wilke et al. used a biaxial cyclic stretching system in 1994 and reported that low-strength stretching facilitates osteoblast proliferation [8]. In 2007, Chen experimented with ultrasound and observed that the growth rate of cells was proportional to the frequency of the ultrasound but was not correlated with the ultrasound's intensity [9]. However, few studies have addressed how cells respond to shear forces (without fluid flowing).

Mechanical stress is an important environmental stimulus for regulating cell physiology and morphology [10–14]. The effects of cyclic stretching, in addition to tensile stress and fluid shear stress, have been investigated in various cell types [15–27]. Regarding how

cyclic stretching impacts cell alignment, Chen et al. reported that the oscillatory force from cyclic stretching tends to shorten the lifetime of trapping bonds, which leads to destabilized focal adhesions [28]. These destabilized focal adhesions then slide or reposition, and the associated stress fibers shorten and rotate into a configuration, which results in an alignment perpendicular to the stretching axis [28,29] It is essential to note that strain amplitude is an important parameter of a cyclic tensile test [30,31]. Excessive strain causes cell death; however, insufficient strain may not cause a response. Different cells have different tensile strain ranges [32–36]. Feng et al. demonstrated that periodic mechanical stimulation, in the presence of IL-1β, increased the expression of MMP, leading to the degradation of the corneal extracellular matrix and the development of keratosis after refractive surgery [37]. Understanding whether corneal cells are affected by mechanical stress can be beneficial, considering the mechanical stresses in daily life. For example, daily intraocular pressure (IOP) causes glaucoma if IOP is too high, and eye rubbing potentially leads to keratoconus. In [37], an analogy is made to the effect of corneal stress after operating refractive surgery. Understanding how mechanical stress affects corneal cells helps physicians to assess the outcomes of corneal procedures in clinical practice. Our experimental setup enhanced the tensional conditions to promote the performance of long-term eye rubbing.

To evaluate the morphological changes that occur during the stretching process, some studies have used finite element analysis (FEA) [38–46] and image recognition [47–50] to determine the mechanical strain characteristics of materials. FEA and other methods of characterizing strain fields are used to determine the biomechanical relationship between cells and mechanical strain. The numerical supports may lead the traditional stretching test to have a strain gradient field through designing structural stress distributions. Thus, a single experiment can obtain several strained results.

This study increased the strain gradient by punching a hole in the center of a stretched membrane to determine the effects of various cyclic strains on cultured human keratocytes (HK, catalog no. 6520, ScienCell, Carlsbad, CA, USA). Two stretching methods were used: cyclic stretching (dynamic), which models eye rubbing, and continuous stretching (static), which models high intraocular pressure. Cyclic stretching was more harmful to cells than continuous stretching, suggesting that eye rubbing does more damage to corneal cells than high intraocular pressure does.

## 2. Materials and Methods

### 2.1. Cell Culture

Experiments were performed using primary HKs (ScienCell, Carlsbad, CA, USA) isolated from human corneas. The cells were cultured in a fibroblast medium containing 1% fibroblast growth supplement (ScienCell, Carlsbad, CA, USA), 2% fetal bovine serum (ScienCell, Carlsbad, CA, USA), and 1% antibiotic solution (P/S, ScienCell, Carlsbad, CA, USA). Fifth-generation HKs were used for the experiments.

### 2.2. Stretching Experiments

These experiments used a square type 1 collagen-coated elastic polydimethylsiloxane (PDMS) membrane (PM22-Collagen I, Taihoya, Taiwan) with an area of 4 cm$^2$ and a hole with a diameter of 3 mm in the center. This commercial membrane is well coated by the collagen and cells could be attached firmly on the PDMS membrane. The 3 mm hole was punched in the center of the membrane to produce a shear strain effect on the membrane after stretching and to increase the strain distribution without damaging the membrane during the stretching process. The PDMS membrane was rinsed with sterile Dulbecco's Phosphate-Buffered Saline (Thermo Fisher, Waltham, MA, USA) and sterilized through exposure to ultraviolet light for 15 min in a biological safety cabinet. A trypsin-treated HK suspension was seeded on the elastic membrane at a density of $1.25 \times 10^4$ cells/cm$^2$ and then placed in 5% CO$_2$ at 37 °C in a carbon dioxide incubator for 24 h. The membrane seeded with HKs was placed in a stretching device designed by the authors' laboratory for uniaxial stretching at a strain of 3% (0.06 mm) for 6 h. Two stretching methods were

investigated: continuous stretching with a constant load and cyclic stretching with a frequency of 1 Hz.

### 2.3. Live Cell Imaging

A reverse-contrast biological microscope (MXK600, MICROTECH, Taiwan) equipped with a charged-coupled device camera (MX-C-MOUNT/0.5X, MICROTECH, Taiwan) was used to capture $5\times$ magnified live cell images every hour. The microscope software MicroCam 5 (M&T OPTICS, Taiwan) was used to capture cell images and to store the image data in a computer. To ensure that the cell images were of the same areas, the stretched sample was used to mark the area to be observed on the bottom of a Petri dish. This mark facilitated alignment and positioning.

### 2.4. Cell Imaging Analysis

Image processing was performed using the library in MATLAB (The MathWorks, Inc, Natick, MA, USA). The images first underwent grayscale processing, top hat replacement, and contrast enhancement, and were subsequently converted to binary images. The binary image conversion comprised the filtering of impurities and filling of holes. Objects caused by noise and other artifacts at the boundary of an image must be filtered out to reduce interference during cell determination. If the contrast inside a cell is different, holes appear in the binarized image. The image area and holes must be filled to accurately depict the cell shape. After the removal of factors that affect cell determination, the location of the cells in an image could be detected and a watershed algorithm could then be used to estimate the number of cells in each image.

### 2.5. Theoretical Analysis of Stress Concentration

To produce a shear strain effect on the membrane after stretching and increase the strain distribution of the membrane, a circular hole was made in the center. As a result of the geometric discontinuity, stress concentrated near the hole. Stress concentration is a phenomenon in which the stress on a part of an object increases. The geometric discontinuity in the membrane leads to an increase in the local stress field. The elastic stress near the circular hole can be calculated using the Kirsch equations:

$$\sigma_{rr} = \frac{\sigma}{2}\left(1 - \frac{c^2}{r^2}\right) + \frac{\sigma}{2}\left(1 + 3\frac{c^4}{r^4} - 4\frac{c^2}{r^2}\right)\cos 2\varnothing, \tag{1}$$

$$\sigma_{\varnothing\varnothing} = \frac{\sigma}{2}\left(1 + \frac{c^2}{r^2}\right) - \frac{\sigma}{2}\left(1 + 3\frac{c^4}{r^4}\right)\cos 2\varnothing, \tag{2}$$

$$\sigma_{r\varnothing} = -\frac{\sigma}{2}\left(1 - 3\frac{c^4}{r^4} + 2\frac{c^2}{r^2}\right)\sin 2\varnothing, \tag{3}$$

where $\sigma_{rr}$ represents the radial stress, $\sigma_{\varnothing\varnothing}$ is the hoop stress, $\sigma_{r\varnothing}$ is the shear stress, $\sigma$ is the uniaxial tensile stress, $c$ is the radius of the hole, $r$ is the radial coordinate (note that $r$ is strictly greater than or equal to $c$), and $\varnothing = 0$ aligns with the loading direction. The Kirsch equations describe elastic stresses around the hole in a plate for one-directional tension. Note that the shear strain is caused by the bending around the hole, which is different from the pure shear designed by a fluid channel, and this type of bending is much closer to the real stresses of tissues, such as stresses induced by rubs and presses.

### 2.6. Finite Element Simulation of Strain Distribution

The finite element software COMSOL was used to simulate the strain distribution of PDMS membranes subjected to periodic stretching. The PDMS membrane was modeled to be the actual size of the test sample and was created using a mesh of triangular elements with automatic meshing. The element type was plane stress, which is governed by the equilibrium equation and Hooke's law for isotropic media. The material parameters of

the model were the PDMS material parameters built into the COSMOL software. The boundary of the model was set with the starting end ($x = 0$) fixed with a predetermined displacement of 0.6 mm (3% tensile strain) in the $x$ direction. Thus, the mechanical behavior of the PDMS membrane was calculated and analyzed, and the strain distribution under periodic stretching was obtained.

*2.7. Statistical Analysis*

All data from independent experiments are expressed as mean $\pm$ standard deviation. The Student's *t*-test was used for statistical analysis by comparing the results of two experiments. Results with $p < 0.05$ were considered to be significant.

**3. Results**

*3.1. Finite Element Simulation of Membrane*

The simulation results of the solid mechanics and steady-state analysis are shown in Figure 1. To verify the stress distribution on the PDMS membrane, a typical geometric model—a circular hole model—was used. A circular hole with a diameter of 3 mm was made in the center of the membrane model. This shape was chosen for two reasons: (1) it is a typical mechanical test model; thus, the strain field is not difficult to obtain; and (2) the area near the circular hole exhibits stress concentration and shear stress. The 3 mm diameter was chosen because the effects of stress concentration on the membrane can easily be observed at this size, yet the hole does not substantially reduce the structural strength of the membrane. Figure 1A is a simulated diagram of the distribution of the normal strain $\varepsilon_{xx}$ on the $x$-axis of an elastic PDMS membrane with a circular hole in the middle. The figure illustrates that the normal strain is at a maximum near the circular hole. Figure 1B depicts the simulated distribution of normal strain, $\varepsilon_{yy}$, on the $y$-axis of the elastic PDMS membrane with a circular hole. As in Figure 1A, the extreme values appear near the circular hole, perpendicular to the stretching direction. The stress concentration effect causes the normal strain near the circular hole to be 5% of the negative direction. Figure 1C shows the simulated shear strain, $\gamma_{xy}$, distribution in the $x$–$y$ plane of the membrane. Most of the area is not affected by shear stress but affected only by tensile stress. The shear stress of the 3% strain acts only on the four corners of the membrane and in the vicinity of the circular hole in the diagonal directions.

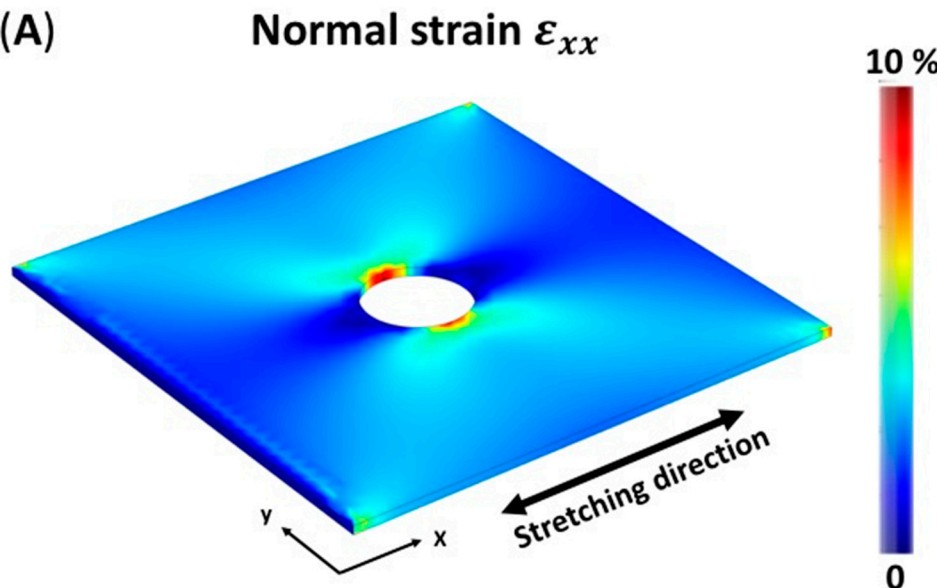

**Figure 1.** *Cont.*

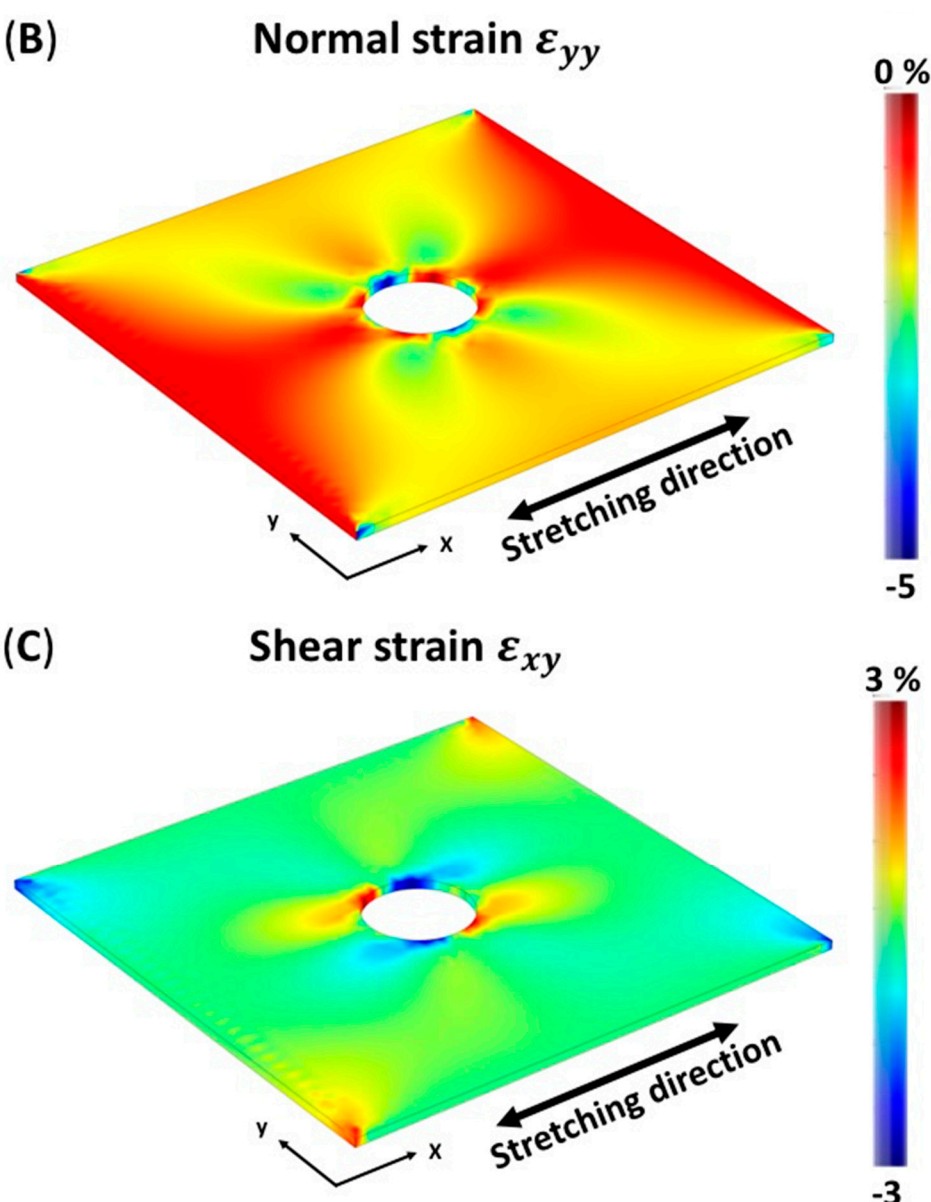

**Figure 1.** Simulation results of the solid mechanics and steady-state analysis. (**A**) Simulated distribution of normal strain, $\varepsilon_{xx}$, on the *x*-axis for the elastic PDMS membrane with a central circular hole. (**B**) Simulated normal strain, $\varepsilon_{yy}$, distribution on the *y*-axis for the elastic PDMS membrane with a central circular hole. (**C**) Simulated shear strain, $\gamma_{xy}$, distribution in the *x*–*y* plane for the elastic PDMS membrane with a central circular hole.

### 3.2. Cyclic Stretching Test Results

HK cells were stretched in the system developed for this study. Cell density and viability were determined by processing captured cell images. For calculating the number of cells, cell staining or examining the color of the medium are widely used methods. However, the color of the medium changes with little precision in the persistent state of cell death. The actual calculation result of the cell number was similar to that of the dyeing result, and it was even more accurate in the local area in our experiment. Moreover, in this study, the sequent changes of the cultured cells were recorded. It was meaningful to actually count the effectiveness and time-dependent changes in cell numbers. On the basis of the simulated FEA, we mapped the strain distribution onto the membrane with implanted cells. At a small distance from the round hole (blue, Figure 2), the strain is

$3 \pm 0.5\%$; closer to the round hole (red, Figure 2), the strain is $10 \pm 1\%$; and areas near the hole corners (orange, Figure 2) undergo shear strain of $3 \pm 0.5\%$.

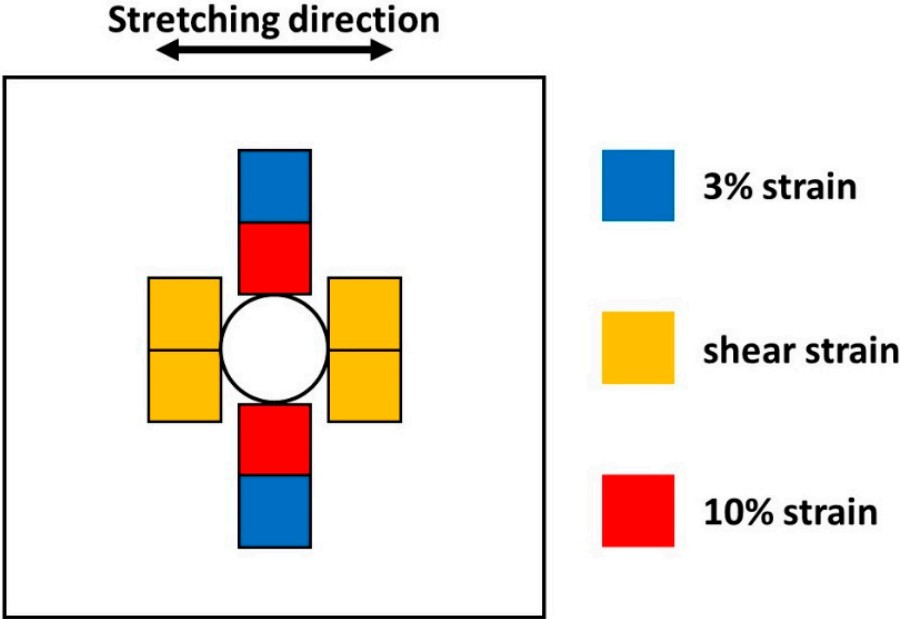

**Figure 2.** Schematic diagram of sampling areas for the stretched sample. Blue: 3% strain. Orange: shear strain. Red: 10% strain.

During the stretching process, microscopic images of the cells were taken. Figure 3 shows a group of cell images before and after cyclic stretching at the same location. These cell images were substituted into the image processing program of MATLAB for quantification and counting. Thus, cell density, apoptosis, cell length and area are typical indicators for understanding post-experimental cell performance. Parameters quantified from microscopic images under cyclic stretching are shown in Figure 4.

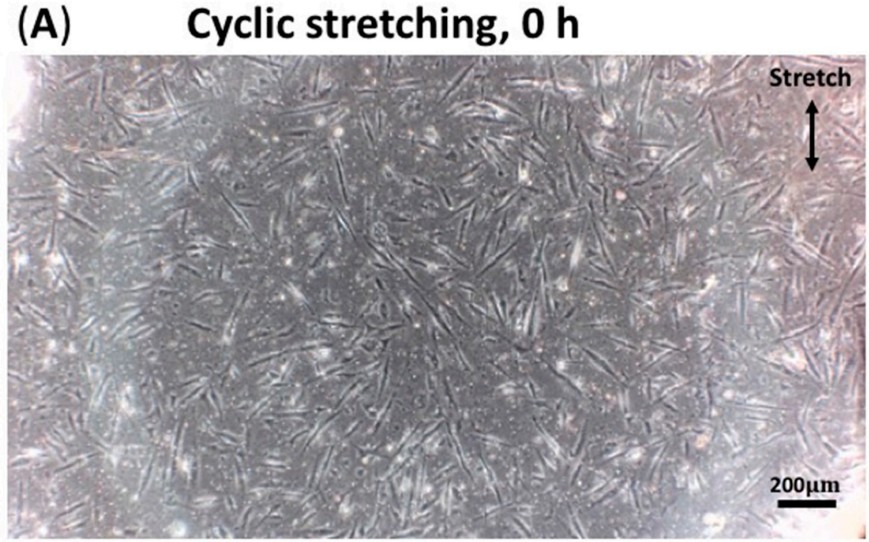

**Figure 3.** *Cont.*

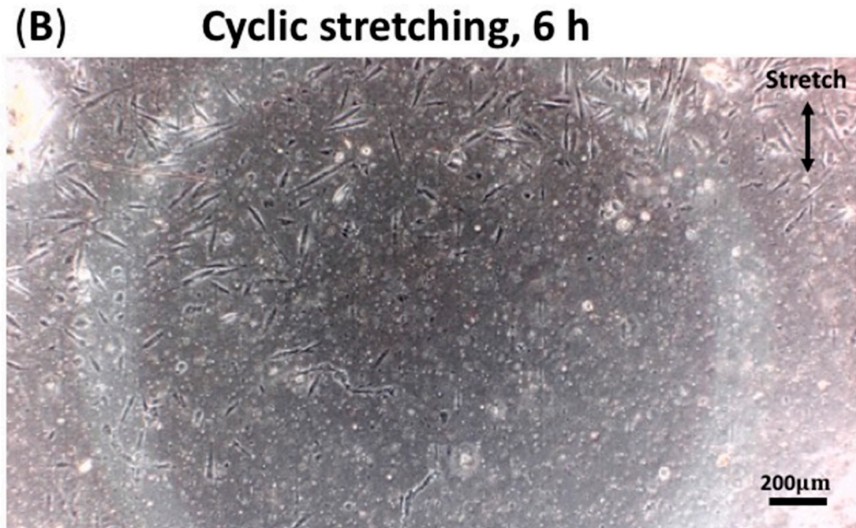

**Figure 3.** Microscopic images of the HK cells before and after cyclic stretching. (**A**) HKs before cyclic stretching. (**B**) HKs after 6 h of cyclic stretching. Scale bar: 200 μm.

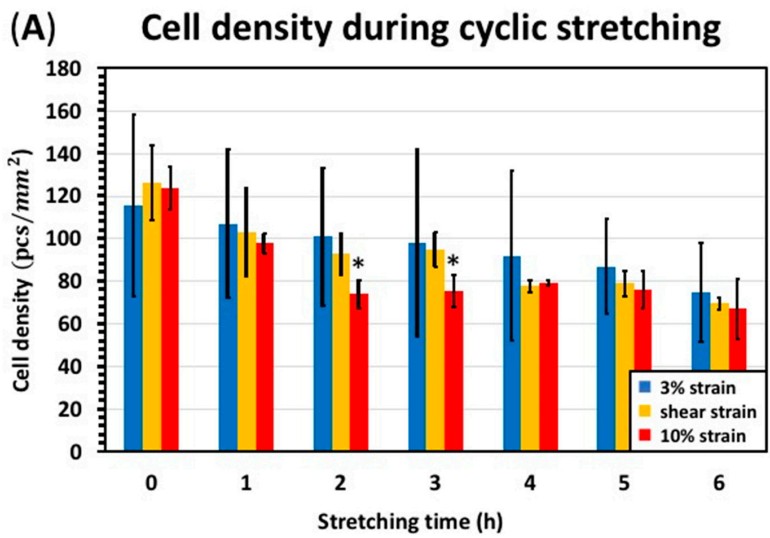

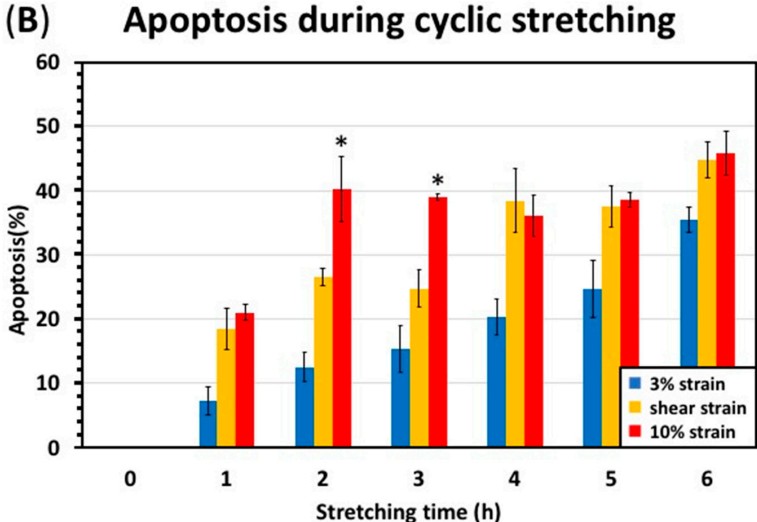

**Figure 4.** *Cont.*

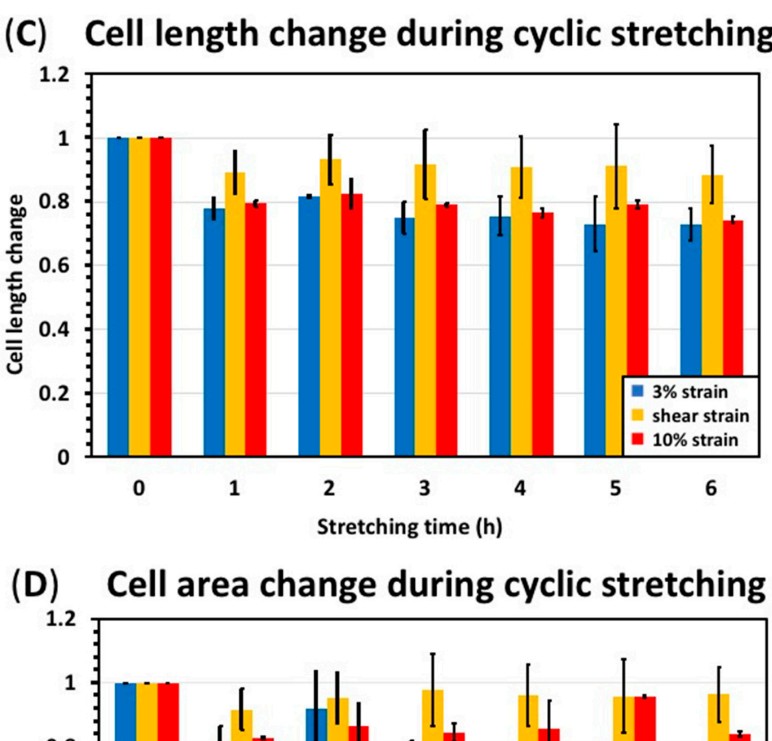

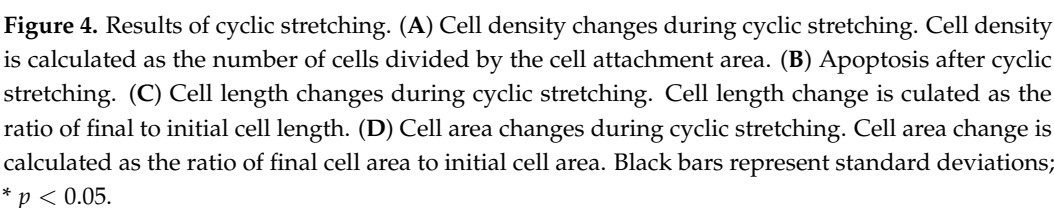

**Figure 4.** Results of cyclic stretching. (**A**) Cell density changes during cyclic stretching. Cell density is calculated as the number of cells divided by the cell attachment area. (**B**) Apoptosis after cyclic stretching. (**C**) Cell length changes during cyclic stretching. Cell length change is culated as the ratio of final to initial cell length. (**D**) Cell area changes during cyclic stretching. Cell area change is calculated as the ratio of final cell area to initial cell area. Black bars represent standard deviations; * $p < 0.05$.

The sample was subjected to 6 h of cyclic stretching strain and 1 Hz. As displayed in Figure 4A, the density of HKs in the 3% strain regions decreased from 115 to 74 pcs/mm$^2$, that in the 10% strain regions decreased from 124 to 67 pcs/mm$^2$, and that in the shear strain regions decreased from 126 to 70 pcs/mm$^2$. Figure 4B displays apoptosis after 6 h of cyclic stretching at 3% strain and 1 Hz. The normalized initial apoptosis was set to 0. The apoptosis of the HKs in the 3%, 10%, and shear regions were 35%, 46%, and 45%, respectively. Figure 4C displays changes in cell length under cyclic stretching. Cell length change is the ratio of the final to the initial cell length; the initial cell length was set to 1. The sample was stretched cyclically at 1 Hz for 6 h. The length ratios of the HKs in the 3%, 10%, and shear regions were 0.78, 0.84, and 0.96, respectively. Figure 4D displays the changes in cell area under cyclic stretching. Cell area change is the ratio of the final to initial cell area; the initial cell area was set to 1. The sample was stretched cyclically at 1 Hz for 6 h. The area ratios of the HKs in the 3%, 10%, and shear regions were 0.73, 0.74, and 0.89, respectively.

### 3.3. Continuous Stretching Test Results

During the continuous stretching, microscopic images of the cells were also taken. Figure 5 is a set of cell images before and after continuous stretching at the same position. These cell images were substituted into MATLAB's image processing program for quantification and counting. Cell density, apoptosis, cell length and area were quantified and derived from these images. Parameters quantified from microscopic images under continuous stretching are shown in Figure 6.

Figure 6A displays the changes in cell density. The sample was stretched cyclically at 1 Hz for 6 h. The density of HKs in the 3% strain region decreased from 89 to 69 pcs/mm$^2$, that in the 10% strain regions decreased from 108 to 70 pcs/mm$^2$, and that in the shear strain regions decreased from 134 to 93 pcs/mm$^2$. Figure 6B displays the apoptosis after continuous stretching. The sample was stretched continuously for 6 h. The apoptosis rates of the HKs in the 3%, 10%, and shear regions were 23%, 35%, and 31%, respectively. Figure 6C displays the changes in cell length after continuous stretching. The sample was stretched for 6 h. The length ratios of the HKs in the 3%, 10%, and shear strain regions were 0.90, 0.87, and 0.87, respectively. Figure 6D displays changes in cell area under cyclic stretching. The sample was stretched cyclically at 1 Hz for 6 h. The area ratios of the HKs in the 3%, 10%, and shear strain regions were 0.82, 0.80, and 0.81, respectively.

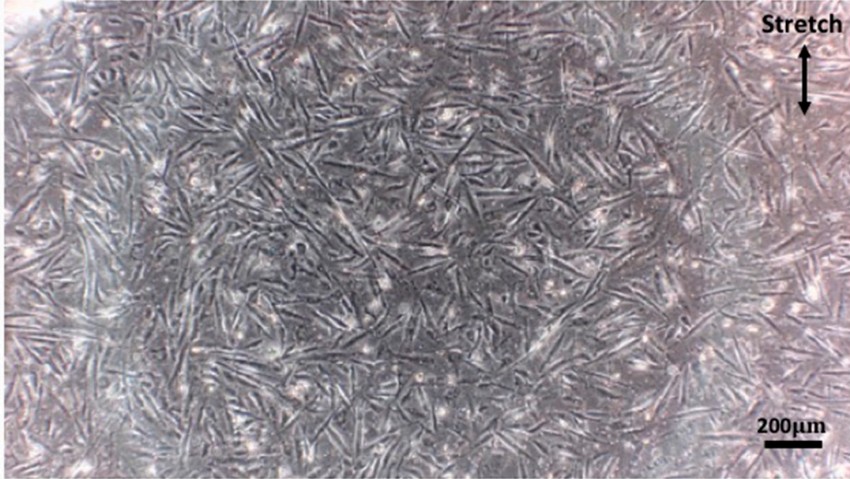

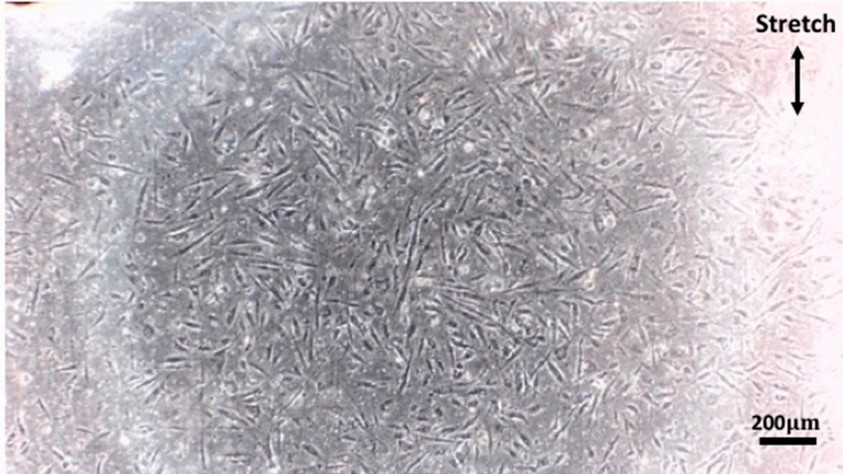

**Figure 5.** Microscopic images of the HK cells before and after stretching. (**A**) HKs before continuous stretching. (**B**) HKs after 6 h of continuous stretching. Scale bar: 200 μm.

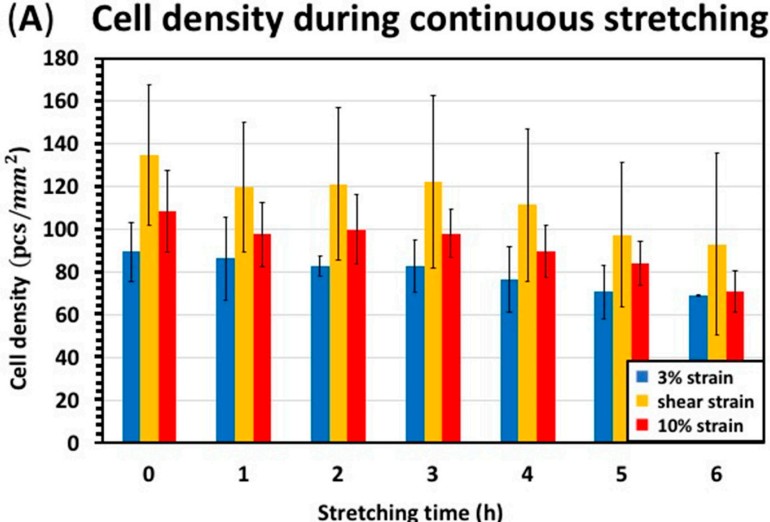

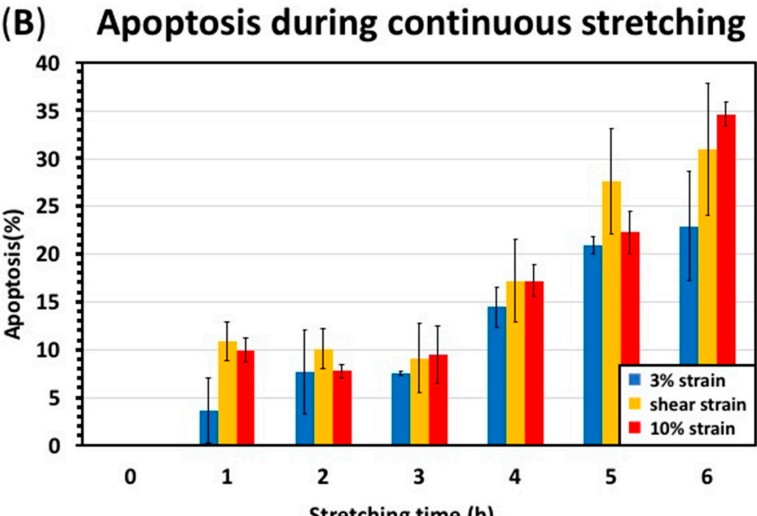

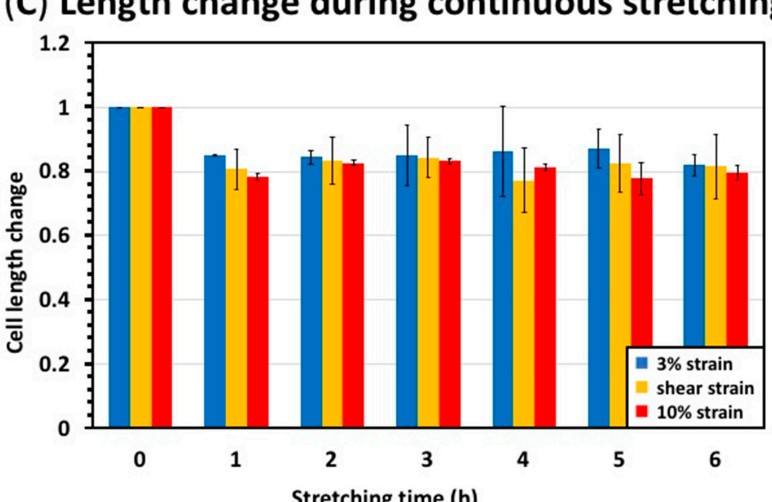

**Figure 6.** *Cont.*

**(D) Area change during continuous stretching**

**Figure 6.** Results of continuous stretching. (**A**) Cell density changes during continuous stretching. Cell density is calculated as the number of cells divided by the cell attachment area. (**B**) Apoptosis after continuous stretching. (**C**) Cell length change during continuous stretching. Cell length change is calculated as the ratio of final cell length to initial cell length. (**D**) Cell area change after continuous stretching. Cell area change is calculated as the ratio of final cell area to initial cell area. Black bars represent standard deviations.

## 4. Discussion

This study aims to propose a method to analyze the behavior of cells under tensile and shear stress in a one-time experiment. HK cells were used in this tensional test since we believe eye rubbing may lead to keratoconus. Ninety percent of the cornea is composed of the stroma, and HK cells are derived from the corneal stroma. Their parallel orientation and arrangement allow the stroma to be transparent, since the distance between the cells is close to half the light wavelength. This study used PDMS films with circular holes in the middle, which increases the strain gradient on the film, allowing us to observe the changes of cells under different strain magnitudes and shear strains simultaneously. This approach is advantageous, as it increases efficiency and drastically reduces the time needed in an experiment that tests different strain magnitudes. In addition, this approach reduces the confounding effect of using separate batches of cells. Due to the limitation of the device and the desire to avoid an overly complicated strain distribution of the film, we choose a single axial stretching direction. Two different stretching methods were used in this study: continuous stretching and cyclic stretching. Continuous stretching is the application of force but no back-and-forth action. Continuous stretching is relatively static but not unstressed. Cyclic stretching is comprised of periodic stretching. In this experiment, the frequency of cyclic stretching was 1 Hz. The duration of both methods of stretching was 6 h. Cell morphology was observed every 1 h, and cell images were taken for quantification.

In this study, high strain and shear induced apoptosis in cells. The strain set on the stretching machine was 3%, and due to the specially shaped film, there were 0–10% strain and shear strain on the film, as shown in Figure 1. We chose 3% (blue, Figure 2) and 10% (red, Figure 2) strain levels to compare the effects of strain on apoptosis. Our previous studies [30,31] reported that HK cells under cyclic stretching reorient in the vertical direction relative to strain. The topics of cell redirection and migration are for separate studies. This study focuses on apoptosis, since cell death could be seen after eye rubbing. After 1 Hz cyclic stretching for 6 h, 23% of cells under 3% strain were apoptotic, which is a lower apoptosis rate than the cells under 10% strain with an apoptosis rate of 36%. After continuous stretching for 6 h, the rate of apoptosis of cells under 3% strain was 35%, which is lower than that of cells under 10% strain, which was 46%. Regardless of the stretching method, the apoptosis rates of cells under 3% strain were lower than those of cells under

10% strain. Furthermore, the magnitude of tensile strain was positively correlated with apoptosis. Additionally, shear strain (orange, Figure 2) induced apoptosis in HK cells. The data presented in Figures 4B and 6B demonstrate an increase in apoptosis after 6 h of shear stress. Therefore, stretching is not conducive to cell proliferation but instead leads to apoptosis.

Two different stretching methods, cyclic stretching and continuous stretching, were used in this study to simulate rubbing conditions and intraocular pressure changes. Cyclic stretching is a cycle of back-and-forth movements. Continuous stretching is relatively static but not unstressed, being the application of a single cycle of protrusion–adhesion–contraction of one cell microtubule, and these stresses are released until all microtubules go into a new protrusion step. Since cyclic stretching is dynamic and intense, cyclic stretching simulates how eye rubbing affects corneal cells. In contrast, continuous stretching is more static and mimics the effects of daily intraocular pressure changes. Figure 7 shows apoptosis after two stretches at different strain regions. The apoptosis rate of HK cells after 6 h of continuous stretching was 23% compared to 35% after 6 h of cyclic stretching (Figure 7A). Figure 7B shows apoptosis under shear strain; after 6 h, the rate of apoptosis of HK cells was 31% for continuous stretching and 45% for cyclic stretching. Figure 7C shows apoptosis at 10% tensile strain; after 6 h, 35% of HK cells were apoptotic after continuous stretching and 46% after cyclic stretching. Therefore, continuous stretching induces less apoptosis than cyclic stretching. Regardless of the strain type, the apoptosis rate after continuous stretching was lower than after cyclic stretching. Therefore, cyclic stretching resulted in apoptosis of keratinocytes. Since cyclic stretching models dynamic perturbation and continuous stretching models elevated IOP, these results suggest that dynamic perturbation damages corneal cells more than elevated IOP does.

The cultured cells were attached as a monolayer on the membrane, so there were no delamination issues in our experiments. From our observations, cells easily attached to the stretched membrane. No cell attachments were found in other areas, such as membrane holders and media tanks. Therefore, cells attaching outside the viewing area was not a problem. Cells in suspension typically do not die within a few hours. However, the cells that failed to attach and were suspended during or after stretching sessions died. We observed that these cells did not attach to the membrane again, even without stretching. Alternatively, it is possible that living cells are detached but not dead, since we used trypsin to detach adherent cells from the membrane to create a cell suspension. Trypsinized cells are alive, but without a place to attach, they cannot survive long in suspension.

Our results after stretching HK cells were compared with other referenced cells under cyclic stretching, as shown in Table 1. In a previous study [51], cyclic stretching could stimulate the growth and arrangement of osteoblasts and thus affect bone remodeling. Initially, we expected our results to be similar to those with stretched osteoblasts. HK cells are one of the main components of the cornea, in which they are arranged directionally. If stretching facilitated the growth and alignment of HK cells, it could help reconstruct corneas and create artificial corneas in vitro. However, after stretching HK cells, no cell growth was seen, contrary to previous studies on other cell types. We tested many conditions, such as the frequency and magnitude strain (0.5–20%), but still saw that cyclic stretching instead facilitated the death of HK cells. We suspected that the cornea was not normally mechanically stimulated like blood vessels and bones. HK cells prefer a stable and undisturbed environment. Stretching HK cells is closer to behaviors, such as rubbing the eyes, that negatively affect the cornea. To summarize, stretching does not help the growth of HK cells but instead results in cell death.

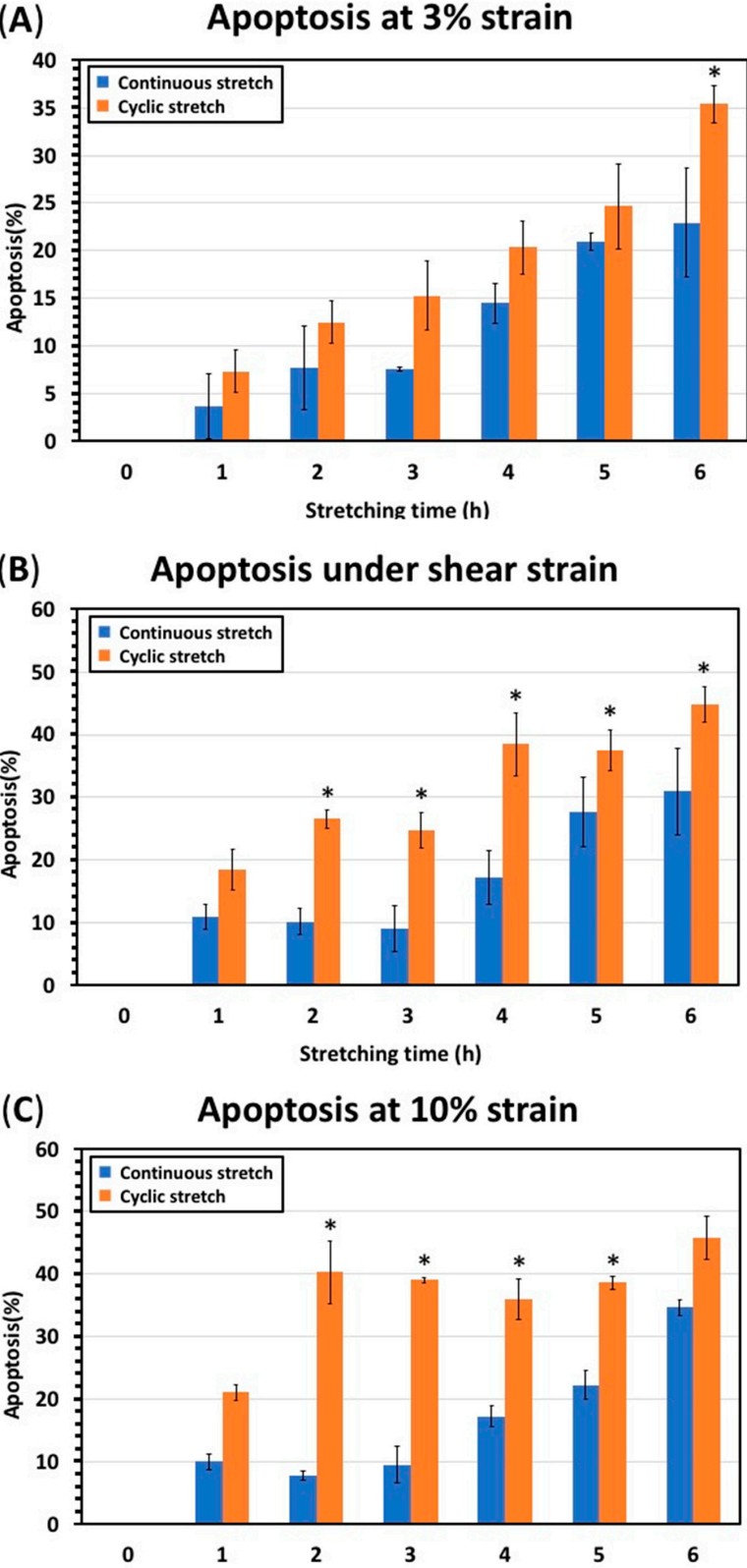

**Figure 7.** Apoptosis after the two types of stretching. (**A**) Apoptosis after continuous and cyclic stretching at a tensile strain of 3%. (**B**) Apoptosis after continuous and cyclic stretching under shear strain. (**C**) Apoptosis after continuous and cyclic stretching at a tensile strain of 10%. Black bars represent standard deviations; * $p < 0.05$.

**Table 1.** Performance of various cell stretching experiments.

| Authors | Cell | Stretch Type | Strain (%) | Frequency (Hz) | Time | Response |
|---|---|---|---|---|---|---|
| Neidlinger-Wilke [8] | Human osteoblasts | Biaxial cyclic stretch | 1.0, 2.4, 5.3, 8.8 | 1 | 15 min per day, 3 days | Proliferation |
| Dai [33] | HBSMC | Uniaxial cyclic stretch | 5, 10, 15, 20 | 0.1 | 6, 12 h | Proliferation |
| Tang [51] | MC3T3-E1 | Uniaxial cyclic stretch | 0, 6, 12, 18 | 0.1 | 24 h | Proliferation |
| This work | HK | Uniaxial cyclic /continuous stretch | 3, 10, Shear | 1 | 6 h | Apoptosis |

## 5. Conclusions

This study has presented a method to study the behavior of cells under tension and shear stress concurrently. Inspired by the classic stress concentration model, we developed a porous stretched membrane. A hole in the center of the stretched membrane changed the strain gradient, allowing simultaneous observation of the effects of stress concentration and shear stress on cells. Using this method, we have shown that both tension and shear stress induce apoptosis in HK cells. The magnitude of tensile strain was positively correlated with the rate of apoptosis. Two stretching methods were used: cyclic stretching (simulating dynamic disturbance of the eye) and continuous stretching (simulating daily IOP changes). Cyclic stretching induced more apoptosis in HK cells than continuous stretching. These findings suggest that dynamic interference may be more damaging to corneal cells than high IOP. Furthermore, the results of these membrane stretching experiments show that cyclic stretching causes apoptosis of HK cells even at low strain.

**Author Contributions:** Conceptualization, J.-Y.Y. and I.-J.W.; methodology, P.-J.S.; software, P.-J.S.; validation, Z.-X.D. and P.-J.S.; formal analysis, J.-Y.Y.; investigation, Z.-X.D.; resources, J.-Y.Y. and I.-J.W.; data curation, Z.-X.D.; writing—original draft preparation, Z.-X.D.; writing—review and editing, P.-J.S.; visualization, Z.-X.D. and P.-J.S.; supervision, I.-J.W.; project administration, J.-Y.Y. All authors have read and agreed to the published version of the manuscript.

**Funding:** This research received no external funding.

**Institutional Review Board Statement:** Not applicable.

**Informed Consent Statement:** Not applicable.

**Data Availability Statement:** Not applicable.

**Acknowledgments:** The authors wish to thank the National Science Council of the Republic of China, Taiwan, for their financial support under contracts MOST-107-2221-E-002-194-MY3, MOST-107-2221-E-011-157-MY3, MOST-107-2221-E-002-186-MY3 and MOST-110-2221-E-011-156-MY2.

**Conflicts of Interest:** The authors declare no conflict of interest.

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
