# Peer review of "Effect of Static and Dynamic Stretching on Corneal Fibroblast Cell"

_processes, doi:10.3390/pr10030605_

Round 1

Reviewer 1 Report

This paper demonstrates increased  strain gradient by punching a hole in the center of a stretched membrane to determine the effects of various cyclic strains on cultured human keratinocytes.

The paper is well written and covers important aspects of the proposed work. Figures are clear with good explanation.

Line 96: "...the database"?

Reviewer 2 Report

The authors have developed a system in which they subject fibroblasts to different mechanical parameters within one experiment. They achieve this by providing a PDMS membrane with a central hole and thereby generating a heterogeneous deformation field including shear. This system is characterized by finite element simulation and used to analyze different parameters of cell behavior under strain. Based on their results, the authors state that ultimately all mechanical parameters have a massive influence on cell survival, cell area and morphology (length change). Here, cyclic strain experiments have an even stronger effect than static strain.  

Although the experimental system could be of potential interest, the authors lack a comprehensive knowledge in cell biology and mechanobiology. This starts with the fact that the authors equate fibroblasts, colloquially called human keratocytes, with keratinocytes, which are epithelial cells and certainly have nothing to do with fibroblasts.
Furthermore, there are a number of other aspects that should have been investigated. For example, there are a large number of publications that investigate the cell response of fibroblasts under cyclic strain. A reorientation in perendicular direction relative to the strain must be expected. At the same time, this should have served as an internal control to verify a stable adhesion of the cells under strain. This is even more important as PDMS membranes are known not to be effectively coated with collagen and thus all results may simply be caused by insufficient adhesion and thereby detachment of the cells from the membrane under strain. It simply does not make sense that all mechanical parameters are cytotoxic and ultimately lead to rounding of the cells. In this case, identical tests would have to be carried out on coatings that are also used for other analyses on PDMS and guarantee a stable cell response (e.g. fibronectin).

Further points:
Reorientation of the cell population was not performed correctly. Here, an analysis must be performed at least in the area unaffected by shear.
Statements that rubbing eyes are dangerous in humans based on simplified in vitro experiments must be removed. 
The material and method section on cell analysis leaves many questions unanswered.

Reviewer 3 Report

Overall a good paper with an interesting approach to characterise strain on cells. The approach with the perforated membrane is useful to check a range of strain parameters in a single experiment. 

The modelling approach is good and clear as well. 

There are a few minor queries:

  • The data are collected for a set of regions or interests for each strain band. it would be good to be clear what range of strain will be experienced in each of these regions (e.g. is the 3% region +-1%?)
  • it would be good to better justify the use of corneal cells. I'm not very expected with this type of cells but would not really expect corneal cells to experience much physical strain (compared to moe active tissues like muscle or bone) Please add clearer rationale for using this type of cells and why the light be exposed to strain (eye rubbing is briefly mentioned but I assume that is unlikely to last for hours) 
  • What will be the benefits from understanding how corneal cell benefits under strain. will this affect clinical practice, is corneal degradation a poorly understood problem?
  • How exactly is 'cell death' assessed? It appears to be just the reduction in number. Where any other markers for cell death used? e.g. live dead staining, apoptosis markers, cytokines, RNA expression etc. Just looking at the number of cells present does not mean all of these cells are still viable. (cells could also be cultured on after the experiment to assess if they have a similar growth rate to a control). Ideally, there should be another measure assessing viability not just visual count.
  • 2d stretch models can also suffer from problems with cell detachment. have the missing cells delaminated but are still viable? (in a 3D model detachment losses are often much lower than in 2D models since there are local high strain points when the cells attach to the membrane. if they attach to other cells in 3D constructs these peak forces are lower so cell loss is often much lower in a 3D construct. Please consider the likeliness of viable cells detaching but not dying, and how these results compare to real organs
  •  

Reviewer 4 Report

In this report, Dai and colleagues have examined the effects of static and dynamic stretching on corneal fibroblast cells. They have used cyclic and continuous stretching regimes to test how such mechanical perturbations affect cellular properties like density, survival, cell size and area. They report minor differences in cell survival rates under some conditions suggesting that these mechanical perturbations may have differential effect on either cell proliferation or apoptosis. Authors however don’t explore this mechanistically and present their results solely as observations. I think the paper doesn’t do a good job of presenting data in context of telling a coherent story, instead presenting a collection of results. Moreover, they need to discuss these results in context of what has been done earlier in the field, this should be done in the discussion.  I have some suggestion and comments for authors to improve their manuscript.

  1. Please consider starting the results from cell experiments with the images presented in Fig 6 ; if these experiments are used for quantification in Fig 3-5. Having the images upfront will allow readers to understand what kind of experiments are being quantified.

  1. The introduction is nice, but discussion needs more work. Please consider comparing your results to what’s done in the field with similar mechanical perturbations.

  1. Also authors have not cited several important literatures in mechanobiology. Please cite the following papers from MP Sheetz and Roca-Cusachs on cell mechanics (PMID : 30403543, 31182865). Also mention that mechanical stretch is sensed and felt at the level of membrane before mechanical signals are relayed to cortical cytoskeleton and please cite following studies (PMID: 31431176, 29632270, 23122885).

Reviewer 5 Report

The manuscript by Dai et al. reports the effect in cell viability and morphology of stretching strain on HK cells.

Although the hypothesis seems of interest for this research, I regret to say I do not see the actual novelty of the work regarding previous works.

The paper has major weaknesses to reconsider before it can be accepted.

The authors use HK cells extracted from the cornea however I am wondering how different, and therefore relevant for the study, are these cells compared to other HK extracted from any other tissue. Why do not use more specific cells with more biological relevance?

Most of the data shown in panels are not statistically significant indicating that the findings are not very meaningful. The author should increase the study including other parameters which might reflect better the effect.

Results and discussion are just considering the presented data and it is not compared or related to previous findings of interest for the work.

The biological discussion is very poor, the authors do not provide any real biologically significance of the data. I would like to see a comparison of the magnitude of the strain induce with real physiological values. This is important for the reader to judge the quality and relevance of the work.

In my opinion, discussing why cyclic strain kills cells and continues does not it would increase the quality of the paper.

Figure 6 should be reconsidered and provide some quantitative data rather than 4 images that one can barely judge if they are representative or not.

The conclusions should reinforced first by building a better a discussion and by highlighting the novelty and questions that the authors want to address.

Round 2

Reviewer 4 Report

Authors have taken the comments of this reviewer into consideration while revising the manuscript.

Reviewer 5 Report

Congrats to the authors for having properly address all my concerns.